# Development of a Method to Locate Deserts and Food Swamps Following the Experience of a Region in Quebec, Canada

**DOI:** 10.3390/ijerph17103359

**Published:** 2020-05-12

**Authors:** Éric Robitaille, Marie-Claude Paquette

**Affiliations:** 1Institut National de Santé Publique du Québec, Montréal, QC H2P 1E2, Canada; marie-claude.paquette@inspq.qc.ca; 2Department of Social and Preventive Medicine, Faculty of Medicine, University of Montréal, Montréal, QC H3C 3J7, Canada; 3Department of Nutrition, University of Montréal, Montréal, QC H3C 3J7, Canada

**Keywords:** food environment, food desert, food swamp, methodology, GIS

## Abstract

Unhealthy eating and food insecurity are recognized risk factors for chronic diseases. Collective and environmental factors, such as geographic access to food condition food choices. The objective of this study was to map food deserts and food swamps in Gaspesie, a region of Quebec (Canada), using geographical information systems (GIS) and field validation. Eleven sectors (5 rural and 6 urban) where 5% of the Gaspesie population lives were considered food deserts. Eight sectors (all rural) constituting 4.5% of the population were considered food swamps. Nearly 88% (3/8) of food swamps were located in disadvantaged and very disadvantaged areas. The Gaspesie region is already actively involved in changing environments to make them conducive to healthy eating for all. The mapping of food deserts can support intersectoral collaboration on food security. Food swamp mapping will make it possible to more accurately characterize the existing food environment in the region. Both indicators will be useful in raising awareness and mobilizing partners for a comprehensive strategy to improve the food environment that is not only based on the food desert indicator alone but also takes into account the presence of food swamps.

## 1. Introduction

Healthy eating is an important determinant of health [1]. Chronic diseases, overweight and unhealthy eating habits, especially when combined with food insecurity, compromise the population’s health [2,3,4]. The high prevalence of these conditions has been shown to generate significant individual, social, and health service costs [5,6]. It is estimated that overweight and obesity alone generate 3 billion CAN$ in direct and indirect health care costs for the province of Quebec [7,8]. Overweight and chronic diseases can be prevented by reducing energy intake and improving the quality of the food offer, particularly among vulnerable populations [9].

The adoption of healthy eating habits depends on individual and collective determinants such as the characteristics of the physical, economic, political, and sociocultural environments [10,11,12]. Improving the population’s diet, therefore, requires a portfolio of multi-targeted and multi-level strategies [13] including those aimed at improving the local food environment.

### 1.1. Local Food Environment and Health

For several years now, Quebec, a province of Canada, has been investing to create healthy eating environments for the population as a whole [14]. Previously, healthy eating was considered to be mainly dependent on individual factors alone. However, collective and environmental factors, including public policies, are now recognized as conditioning food choices by making them easier to make [10].

The population’s diet is influenced by the food environment to which it is exposed [8]. In particular, diet is associated with the community or local food environment, which refers to the characteristics of the places where food can be obtained, i.e., the type of outlet, its location, number of shops in a sector, and geographical accessibility [15]. The local food environment includes businesses that offer food products for retail sale (convenience stores, grocery stores and supermarkets), restaurants (with table service or quick service) and alternative forms of food distribution (farmers’ markets, short food supply chain, online shopping, etc.) [15,16].

Accessibility is a key concept when it comes to the local food environment. It encompasses commercial availability, geographic access, convenience, economic access, and social and cultural acceptability. This paper focuses exclusively on geographical accessibility and its influence on the location of food deserts and food swamps [16,17,18].

Several studies have shown that the availability and accessibility of healthy food can vary according to the income and educational level of individuals living in an area [19,20]. These disparities in access to food may in turn be related to eating habits and behaviours or obesity [17,20,21]. Access to retailers offering healthy food in socioeconomically disadvantaged neighbourhoods is more difficult than elsewhere. This prompted the studies on food deserts. The term food desert refers to an area that is highly disadvantaged socioeconomically and provides poor or low access to food stores offering food of high nutritional value in urban, non-metropolitan and rural areas [22,23]. It was originally used in studies from the United Kingdom [19].

Other studies in the United States and Canada have also noted that socioeconomically disadvantaged neighbourhoods are characterized by a concentration of fast food restaurants and convenience stores. These neighbourhoods can be considered as food swamps [24,25,26,27,28]. Food swamps are food environments where access to food of low nutritional value is so important that it “drowns” access to healthy food of high nutritional value [29]. There is no single definition or indicator of food deserts or food swamps. This explains, for example, why some researchers consider food swamps to be located only in disadvantaged areas [27], while others do not make this distinction [30].

Research has shown an association between body weight and food swamps [20,26,31,32]. The results of a recent study conducted in the United States in 3141 counties shows that even after statistically controlling for the effects of food deserts, food swamps still had a significant positive effect on the rate of obesity in adults [33]. In this study, the regression analyses showed that the association of food deserts to obesity became statistically insignificant when the food swamp indicator was included [33]. These researchers concluded by recommending that policy-makers address both food deserts and food swamps to improve the population’s eating habits and body weight [33]. Furthermore, Canadian researchers have hypothesized that in Canada, where food deserts do not seem as widespread as in the USA, food swamps may be a more salient indicator of food environments [26].

### 1.2. Study Objectives

This paper has two objectives. The first objective is to illustrate, using a method based on geographical information systems (GIS) and field validation, the development of indicators to identify sectors qualified as food deserts and food swamps for the region. Based on the results of this approach, the second is to present how the regional public health authority (RPHA) disseminated food deserts and food swamps mapping to partners and used this information to stimulate regional food environment actions.

## 2. Materials and Methods

### 2.1. Data Sources

Four existing governmental data sets were used to operationalize the indicators on food deserts and food swamps: (1) the location of businesses where food is sold, (2) a disadvantage index, (3) the spatial distribution of residential units, and (4) the geometry of the Canadian census. The Gaspesie-Iles-de-la-Madeleine (shortened to Gaspesie in this article) region, one of the 18 health authority regions of Quebec, was chosen for this study. This eastern region of Quebec is composed of coastal shorelines and an archipelago of islands in the St-Lawrence Gulf. The total population of the region is around 90,000, which is distributed in city centres surrounded by large rural areas.

The ministère de l’Agriculture, des Pêcheries et de l’Alimentation du Québec (MAPAQ)’s (Agriculture, Fisheries and Food of Québec) 2018 registry database of food sales permits provided information on the location of food businesses [34]. This registry collects information on various retail permits which include the following categories: supermarkets, grocery store and convenience store, and, fast-food restaurants [34]. Unlike data from commercial directories, registration in the food sales permit registry is mandatory.

The disadvantage index used was developed at the Institut national de santé publique du Québec (INSPQ) (National Public Health Institute) [35]. It identifies sectors that are socio-economically disadvantaged and is based on 2011 data from Statistics Canada’s National Household Survey [36]. For this analysis, the sectors considered disadvantaged are those in the 5th quintile (Q5) on the material dimension of the deprivation index [35]. This includes about 20% of the population with the least favourable material conditions.

Information on the spatial distribution of residential units came from the 2016 building location file of the ministère des Affaires municipales et de l’Occupation du territoire (Municipal Affairs and Housing) [37].

The geometry of the 2011 Canadian Census was used to distinguish between rural and urban areas. Urban areas were those located in population centres that are areas with a population concentration of at least 1000 people and a population density of at least 400 people per square kilometre [38].

### 2.2. Categorization of Stores Where Food Is Sold

#### 2.2.1. Food Stores

Generally, supermarkets, grocery stores, and fruit and vegetable stores are considered to be food stores that offer foods of high nutritional value [39]. In the retail permit registry, supermarkets are defined as sales establishments, with a much larger surface area than conventional grocery stores. They have an area of more than 2500 square m. As for grocery stores and fruit and vegetable shops, the registry defines them as “establishments ranging in size from 400 to 2500 square m, where food of all kinds is sold (…)” [34].

#### 2.2.2. Fast Food

As for fast food restaurants, the retail permit registry defines them as “establishments where light meals, whether or not consumed on-site, are prepared and served”, “whose main activity is the preparation and sale of food to be taken away or delivered” or “with counter service and occasionally at the table, whose main activity is characterized by the preparation of specialized menus: hamburgers, chickens, hotdogs, etc. (…)”. Fast food restaurants are considered to offer foods of low nutritional value.

#### 2.2.3. Convenience Stores

Convenience stores with and without gasoline sales were grouped in our typology. Most “convenience store” type businesses were grouped in the MAPAQ database under the category of grocery, convenience store, or fruit and vegetable store permit. An automated search for the various “convenience” stores was carried out. The automatic process consisted of searching through certain keywords (i.e., convenience stores, accommodation) or company names (i.e., Proprio, Boni-Soir, Couche-Tard) in the various fields of the database. Convenience stores are considered offering foods of high caloric density and low nutritional value.

### 2.3. Geolocation of Shops and Field Validation

A batch geolocation service provided by the ministère de la Sécurité publique du Québec (Ministry of Public Security) (2018) was used to geocode the various food stores in the regions [40]. Regional public health professionals validated the information contained in the registry by checking if existing stores were still in operation and by adding food stores that were not necessarily included in the registry but which they believed contributed to the local food environment. For instance, the region’s public health professionals decided to add fishmongers and butcher shops as food stores. Moreover, while most studies classify convenience stores as offering unhealthy foods, their offering can differ in remote rural areas. In some instances, public health professionals considered them as small grocery stores.

The geolocation process identified 64 food stores (supermarkets, grocery stores, fruit and vegetable stores), 76 convenience stores, and 108 fast food restaurants. After the field validation process, 41 food stores (fishmongers, butcher shops and convenience stores considered as grocery stores) were added, 10 convenience stores that were considered as small grocery stores were deleted and no fast-food restaurants were added or deleted (see Table 1).

### 2.4. Operationalization of the Food Desert Indicator

The calculation of spatial accessibility to food stores is based on the distance to food stores in metres, calculated from the road network (reticular distance). Calculations were done using ArcGIS 10.5 software and the “network analyst” function [41]. The reticular distance is calculated between the origin which is represented by the residential property assessment units and the nearest business. A common measure in the accessibility of proximity services studies [42] (see Figure 1):$$A=min\left|d_{ij}\right|$$
whereA = distance between residential units i and food store j.

One of the aspects to be considered when developing a food desert indicator is the determination of a low access threshold. There are many definitions of a low access threshold in the scientific literature, with distances ranging from 450 m to 1600 m in urban areas [17,43,44].

The United States Department of Agriculture (USDA) in a mapping tool for locating food deserts in the United States uses the 1 mile (1600 m) threshold in urban areas and 10 miles (16,000 m) in rural areas (USDA, 2018) [45]. In other words, areas with supermarkets located more than 1 mile or 10 miles away are considered to have low access to these types of businesses. The 1000-metre threshold in urban areas is used in this research because it is generally agreed that an adult can walk this distance in less than 15 min. A food store located at this distance is therefore accessible to those who do not have access to a motor vehicle, a well-developed public transit system, and to people able to walk this distance.

In rural areas in the USA, Blanchard and Matthews (2007) and the USDA used a distance of 16,000 m [22]. The low access threshold used in the scientific literature is generally higher in rural areas since it is agreed that motor vehicle ownership in these areas is higher. Nevertheless, using this threshold does not reflect the reality of some population groups such as persons without a car or seniors that cannot drive. Furthermore, people without access to a motor vehicle in rural areas may have greater difficulty accessing food than in urban areas, since the public transit system is often non-existent or less developed in these areas.

In short, food deserts are areas in the 5th quintile of the material deprivation index where access to food stores (supermarkets, grocery stores, fruit and vegetable stores) is low, that is, where the population is on average more than 1000 m from a food store in urban areas and more than 16,000 m in rural areas.

### 2.5. Operationalization of Areas Qualified as Food Swamps

To map the areas that could be qualified as food swamps, the Retail Food Environment Index (RFEI) was calculated for each residential property assessment unit by dividing the number of stores that were less conducive to healthy eating (fast food outlets and convenience stores), by the number of stores categorized as offering healthy food (supermarkets, grocery stores, fruit and vegetable shops) in a 1000 m reticular buffer zone around residential units in urban areas and 16,000 m around those located in rural areas. Subsequently, data at the scale of residential property assessment units were aggregated at the scale of dissemination areas.
$$\mathrm{RFEI}=\frac{\left(fast\;food+\mathrm{convenience}\;\mathrm{stores}\right)}{food\;stores\;\left(supermarket\;and\;grocery\;stores\right)}$$

It has been shown that RFEI is associated with a high density of stores [46,47], an area with an RFEI higher than 5.0 is associated with higher risks of diabetes and obesity prevalence. For this study, food swamps were defined as areas with an average index of at least 5.0; that is, where there are on average 5 times more stores that are less supportive of healthy eating than there are stores that are supportive of healthy eating (see Figure 2).

## 3. Results

### 3.1. Food Deserts

The results show that in Gaspesie, 11 sectors (dissemination areas), 5 in rural areas, and 6 in urban areas, are considered to be food deserts. This means that in these areas of high material deprivation, for the residential property assessment unit, it is necessary to travel more than 1000 m in urban areas and more than 16,000 m in rural areas to reach the nearest food trade. According to 2016 Canadian Census data [48], 4499 people, which represent 5% of the region’s population, live in a food desert. Figure 3 illustrates the spatial distribution of areas referred to as food deserts and the level of geographical access to food stores (Figure 3).

### 3.2. RFEI and Food Swamps

The mean RFEI index for the region is 1.76. On average, there are approximately seven fast-food restaurants and convenience stores for every four supermarkets and grocery stores. There does not seem to be a trend in RFEI according to area deprivation status. However, this is not the case for food swamps. More food swamps were found in areas with higher levels of deprivation compared to areas with lower levels of deprivation.

Concerning food swamps, in Gaspesie, eight sectors (dissemination areas), all in rural areas and none in urban areas, are considered food swamps (Table 2). In these sectors, the RFEI is above 5.0. This means that there are five convenience or fast food stores for each supermarket, grocery, or fruit and vegetable store in these areas. In this region, all food swamps are located in rural areas and nearly 88% (3/8) are located in disadvantaged or very disadvantaged areas (quintiles 4 and 5). It is also in these sectors that the RFEI is the highest. In urban areas, no food swamps were identified. According to the 2016 Canadian Census data [48], 4128 people live in a food swamp, representing 4.5% of the region’s population. Figure 4 shows the spatial distribution of the areas referred to as food swamps and the RFEI (Figure 4). No area is both desert and swamp.

### 3.3. Examples of Regional Actions and Policy Implications Following Local Food Environment Mapping

Residential areas corresponding to the criteria of food deserts and food swamps were mapped in urban and rural areas of the region. These maps reflect the geographical accessibility to food retailers in a region. They are complementary to regional expertise and observations and can be used as a basis to generate reflection and action on the local food environment.

The food desert maps can support several actors in their reflection on food deserts and on initiatives that need to be considered to improve food security and access to healthy food. Intersectoral regional instances on poverty can use the food desert maps in their work to decrease food insecurity as decision support tools. Some RPHA professionals could also use the food desert maps to build policy briefs for regional governing bodies and raise awareness and promote initiatives that increase access to healthy foods such as the opening of a farmers’ market or in a meeting on agricultural zoning development plans.

In the Gaspesie region, the food desert maps were used as key information for the development of a food system approach that translates to “Feeding our people; A collective reflection on food autonomy” [49].

## 4. Discussion

The first objective of this study was to develop indicators of food deserts and food swamps to be used by the Gaspesie region. It also aims to present how regional public health authorities can use these indicators and food environment mapping to raise awareness, mobilize and support their partners to act on the characteristics of the region’s food environment to increase food access and reduce food insecurity.

Analysis of the Gaspesie food landscape revealed that 11 sectors, both urban and rural, are characterized by poor access to stores offering healthy food. This represents 5% of the region’s population. Also, eight sectors (4.5% of the region’s population) are qualified as food swamps meaning that in these sectors foods of low nutritional value are prominent. An important proportion of residents (88% (3/8)) that are exposed to food swamps live in disadvantaged areas. This means that about 10% of the region’s residents are exposed to a food environment that is not conducive to healthy eating.

Research on food swamps is in its infancy; however, some authors suggest that these environments have particularly pernicious effects by encouraging people to make poorer food choices since they are made quickly to experience immediate gratification [50]. Bridle-Fitzpatrick (2015) [51] notes that participants in her study who were continuously exposed to foods and beverages of low nutritional value reported that they now perceive these products as normal and that their desire to consume them was therefore increased. In their study, exposure to food swamps appeared to influence not only food choices, but also food preferences and norms. This author, therefore, considers that actions on food deserts are necessary, but insufficient to “clean up” food swamps, which requires more robust interventions than increasing access to healthy food for disadvantaged populations [51].

It has been suggested that in urban areas in Canada, food swamps may be a more salient metaphor to characterize the food environment than food deserts [26]. This study shows that in a rural setting, food deserts and food swamps co-exist. Still, more research needs to be done to elucidate the associations between the indicators of the food environment, eating behaviours, food insecurity, and indicators of health.

Public health agencies and researchers recognize the importance of developing and strengthening the local food environment to make it favourable to healthy eating and supportive of food security [52,53,54,55]. Promising interventions to improve physical access to food can be divided into four categories: introducing new conventional sources of supply (e.g., supermarkets) [56,57,58] or alternative sources of supply (e.g., solidarity grocery stores, public markets, and mobile markets) [59,60,61], improving in-store food offering (e.g., fruits and vegetables in convenience stores) [62,63], land use planning (e.g., zoning) [64], and, increasing mobility (transportation infrastructure) [63].

In 2014, the CDC published a guide outlining strategies to be carried out to improve physical access to more nutritious food [65]. Before putting into place promising interventions such as those outlined above, they propose to initially evaluate the characteristics of the food environment. The methods illustrated in our research could facilitate this evaluation of the local food environment. For example, in the United States, interventions to increase healthy food access often take the form of tax exemption, financial incentives or zoning by-law amendments to encourage entrepreneurs to open a new grocery store or supermarket in communities designated as food deserts [66]. Notably, in Philadelphia in recent years, the Pennsylvania Fresh Food Financing Initiative (public-private partnership) has funded the establishment of many food stores and cooperatives [67]. Urban or community-supported agriculture could also improve the situation of people living within areas known as deserts and food swamps. This is the conclusion of the analyses of urban agriculture initiatives in New York and Detroit. Urban agriculture becoming the starting point of a food system by providing, for example, produce to local markets [68,69].

The Gaspesie region has also put into place interventions to improve access and reduce food insecurity. Notably, they invested in increasing mobility to facilitate access to existing shops by providing buses that go to local stores. This type of intervention could also be beneficial in influencing food swamps since a recent study reported a greater association between food swamps and obesity in areas where the population had lower mobility [33]. Improving population mobility reduces the impact of food deserts and food swamps on the population.

The food swamp map of Gaspesie could also be used to implement measures that would limit the introduction of new fast food or convenience stores in areas identified as being at risk because of the higher density of these types of businesses compared to supermarkets, grocery stores and fruit and vegetable stores [29].

The food desert and food swamp data reported in this study should be interpreted with caution. Several limitations are associated with the methods used. First, there is no universal measure of deserts and food swamps, nor is there consensus on the thresholds to be used that are associated with consequences on food quality or population health. Second, the categorization of food stores based on MAPAQ’s registry is not optimal, as some stores categorization could not accurately reflect the products offered and new stores could have opened that would not be included in the registry. This limitation is often raised in the literature [70,71]. We have addressed this limitation by using field validation with public health professionals to improve the validity of the MAPAQ registry.

Furthermore, the RPHA decided to add fishmongers to the healthy stores category as well as fruit and vegetable shops. These stores were not initially included in the indicator as it was deemed that they could not provide the wide range of foods that are necessary for healthy eating. In total the region added 41 food stores and deleted 10 convenience stores. These changes to the database in effect “decreased” the number of food deserts and of food swamps than would have been identified if using only the MAPAQ registry.

An additional limitation is the variable offering found in some food stores. For example, while supermarkets qualify as stores offering food of high nutritional value they also offer a wide range of foods of low nutritional value [72].

Finally, physical accessibility is only one way of describing the food environment. Some argue that the concepts of deserts or food swamps place too much emphasis on spatial distribution to the detriment of other dimensions of accessibility such as income, household characteristics, transportation opportunities and the notion of (available) time, which are also factors that need to be addressed because they are closely related to eating behaviours [29,73].

## 5. Conclusions

This study has shown the presence of food deserts and food swamps in Gaspesie, a region located in Canada’s province of Quebec. The mapping of food environments represents an interesting tool for mobilization and reflection to implement interventions that can promote the creation and development of environments more conducive to healthy lifestyles [74,75].

In recent years, the Gaspesie region used the mapping of food deserts to support the discussions with their partners. The addition of a food swamp mapping will allow them to more accurately characterize the existing food environment in the region. It will be useful in raising awareness and mobilizing partners for a comprehensive strategy to improve the food environment that is not only based on the desert indicator alone, but also takes into account the presence of food swamps.

## Figures and Tables

**Figure 1 ijerph-17-03359-f001:**
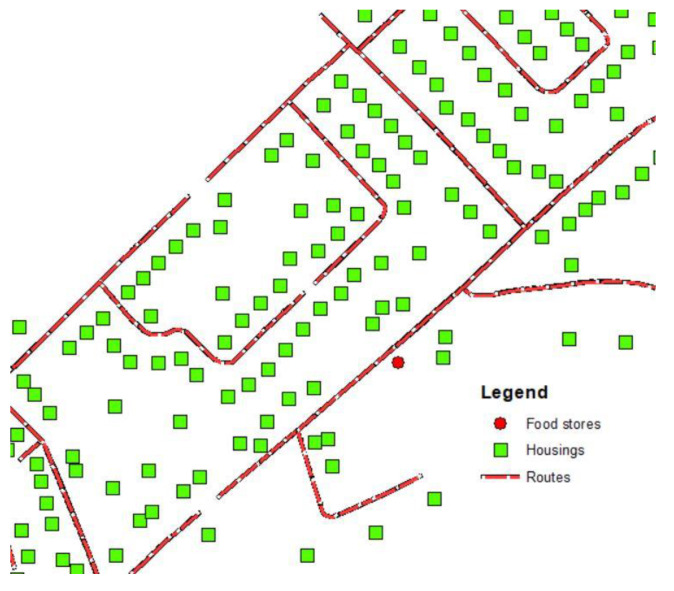
Food desert store accessibility calculation.

**Figure 2 ijerph-17-03359-f002:**
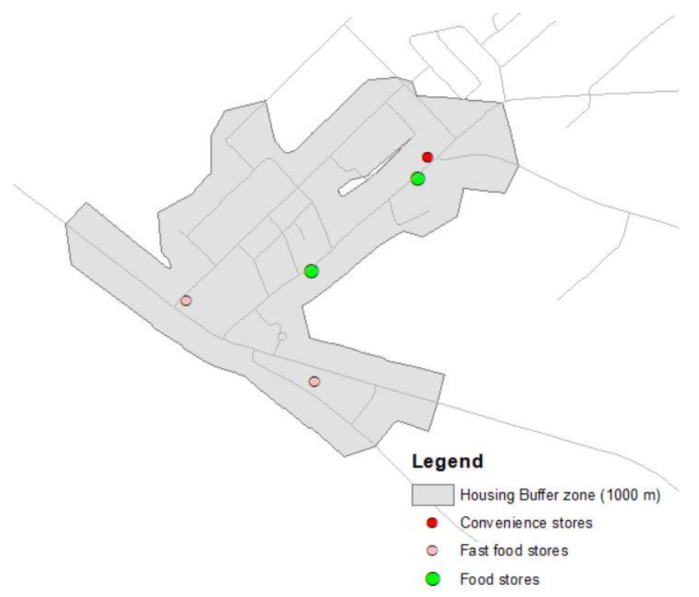
RFEI calculation method.

**Figure 3 ijerph-17-03359-f003:**
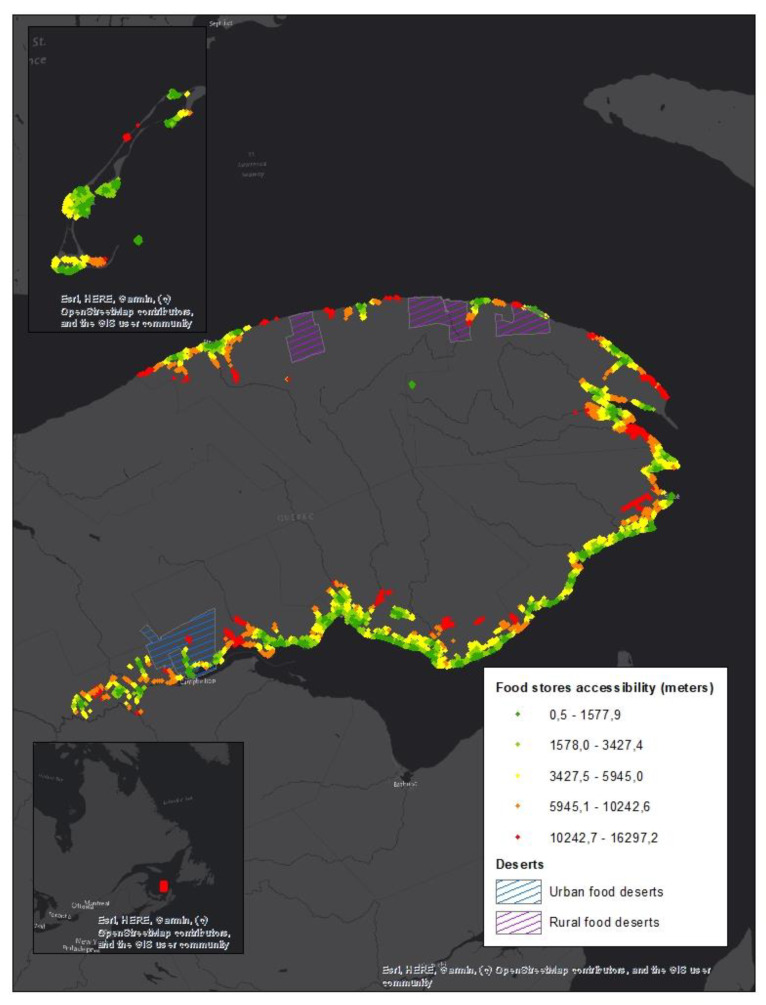
Food desert localization.

**Figure 4 ijerph-17-03359-f004:**
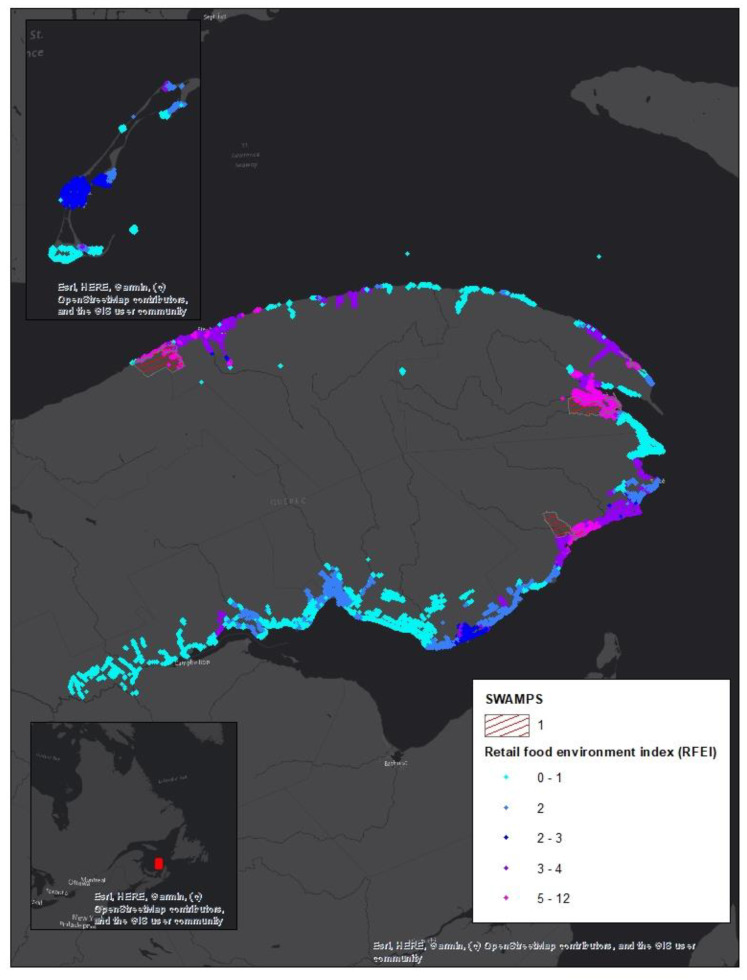
Swamp area localization.

**Table 1 ijerph-17-03359-t001:** Number of food stores before and after the field validation process.

Before Field Validation			After Field Validation and Use in Analysis		
# Geolocated Food Stores	# Geolocated of Convenience Store	# Geolocated of Fast Foods	# Geolocated Food Stores	# Geolocated of Convenience Store	# Geolocated of Fast Foods
64	76	108	105	66	108

**Table 2 ijerph-17-03359-t002:** Average level of the RFEI and number of sectors qualified as food swamps by disadvantage quintiles of RFEI # of sectors in rural areas # of sectors in urban areas.

Quintiles of Disadvantage Average Level	Mean Level of RFEI	# of Sectors in Rural Areas	# of Sectors in Urban Areas	Total
No data	1.04	0	0	0
1 (less deprived)	1.14	0	0	0
2	2.52	0	0	0
3	2.11	1	0	1
4	2.36	4	0	4
5 (more deprived)	2.22	3	0	3
Total	1.76	8	0	8

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
