# Peer review of "Development of a Method to Locate Deserts and Food Swamps Following the Experience of a Region in Quebec, Canada"

_ijerph, 2020, doi:10.3390/ijerph17103359_

Round 1
Reviewer 1 Report
The authors provided an interesting paper on geolocation of food deserts and food swamps and came up with useful correlations. Although this is a case-specific study (Quebec, Canada), the approach is well presented.
Here are some additional comments:
1. Figure captions should be below the figure.
2. The used font in the legend figures is quite small. I propose to increase to have more visible information
3. The "introduction" paragraph is quite brief. The authors should provide some more background related to their study.
4. Table 1, should be placed within one page. At the current version, it is merged between two pages.
5. Equations and formulas should be aligned in the center.
Author Response
Reviewer 1
The authors provided an interesting paper on geolocation of food deserts and food swamps and came up with useful correlations. Although this is a case-specific study (Quebec, Canada), the approach is well presented.
Response: Thank you for your comments.
Here are some additional comments:
- Figure captions should be below the figure.
Response: We made the corrections.
- The used font in the legend figures is quite small. I propose to increase to have more visible information
Response: We made the corrections.
- The "introduction" paragraph is quite brief. The authors should provide some more background related to their study.
Response: We added information in the introductory paragraph.
- Table 1, should be placed within one page. At the current version, it is merged between two pages.
Response: We made the corrections.
- Equations and formulas should be aligned in the center.
Response: We made the corrections.
Reviewer 2 Report
I have reviewed a manuscript entitled ' Development of a method to locate food deserts and food swamps following the experience of a region in Quebec, Canada' . The manuscript has clear distinct research objectives which have been clearly designed at the end of the introduction section.
Based on the nature of the study conducted, this manuscript is making an important theoretical advancement as the research on food deserts and food swamps is relatively new and these concepts can be meaningfully correlated with population health aspects, and especially the likelihood of disease. As a result, the research is casting new light on the role of collective and environmental factors such as how accessibility to different types of food stores that determine food choices in both rural and urban areas.
The investigative methodology using GIS have been satisfactorily described and are appropriate to the research. The primary data collected has been accurately and objectively interpreted while the results are clearly presented and subsequently discussed. The implications for appropriate policy development and interventions are drawn from these results. it can be stated that the significance of these results has been adequately demonstrated, along with an explanation on the limits of this research and what possible new strands of research are being recommended.
While the composition and compilation of this manuscripts is of a high standard, there are minor grammatical flaws here and there. These areas have indicated by comments made of the pdf manuscript. If these language writing problems can be duly fine tuned, my recommendation is that this manuscript is publishable.
Detailed comments please find attached.

Author Response
Reviewer 2
I have reviewed a manuscript entitled ' Development of a method to locate food deserts and food swamps following the experience of a region in Quebec, Canada' . The manuscript has clear distinct research objectives which have been clearly designed at the end of the introduction section.
Based on the nature of the study conducted, this manuscript is making an important theoretical advancement as the research on food deserts and food swamps is relatively new and these concepts can be meaningfully correlated with population health aspects, and especially the likelihood of disease. As a result, the research is casting new light on the role of collective and environmental factors such as how accessibility to different types of food stores that determine food choices in both rural and urban areas.
The investigative methodology using GIS have been satisfactorily described and are appropriate to the research. The primary data collected has been accurately and objectively interpreted while the results are clearly presented and subsequently discussed. The implications for appropriate policy development and interventions are drawn from these results. it can be stated that the significance of these results has been adequately demonstrated, along with an explanation on the limits of this research and what possible new strands of research are being recommended.
While the composition and compilation of this manuscripts is of a high standard, there are minor grammatical flaws here and there. These areas have indicated by comments made of the pdf manuscript. If these language writing problems can be duly fine tuned, my recommendation is that this manuscript is publishable.
Response: Thank you for your comments.
Detailed comments please find attached.
- This sentence looks okay but can be expanded just a little bit for a greater and effective finish off . So here is my example below: This paper focuses exclusively on geographical accessibility and its influence in the location of food deserts and food swamps [10–12].
Response: We changed the sentence for the proposal.
- I suggest that you add the word 'or' between 'poor' and 'low'. In that way, you are more discreet about what you mean unlike leaving the reader to decide
Response: We changed the sentence for the proposal.
- You have two choices here: (1) ......a more salient indicator of a food environment [20] or (2) ...a more salient indicator of food environments [20]. Furthermore, pls make sure that you leave one empty space between the word 'environment' and the [20].
Response: We changed the sentence for the proposal.
- While I appreciate the Quebec province's French basis, this journal caters for research work expressed in the English language. So, can you pls provide the English equivalent of all the French words in this line and elsewhere in this manuscript . Such English equivalents must be placed inside Brackets '( )' . For example, see below and complete the translation.
The ministère de l'Agriculture, des Pêcheries et de l'Alimentation du Québec (MAPAQ)’ (Ministry of Agriculture; etc etc ) 2015 registry database of food sales permits provided information on the location of food businesses [28].
Response: We changed the sentence for the proposal as well as any other French names in the manuscript.
- What is the purpose of these two semi-arrows in front of the full stop? If there is no purpose can you please delete them.
Response: We changed the sentence for the proposal.
- Usually the captions or labels of figures in a scientific manuscript are provided below such figures but where tables are given the manuscript, the caption is usually provided on top of such an illustration.
Response: We made the corrections.
- a region located in Canada's province of Quebec.
Response: We changed the sentence for the proposal.
Response: All comments made by this reviewer in the pdf file was modified in the final manuscript. Thank you!
Reviewer 3 Report
Food deserts are characteristic of the United States and for several years has become a social problem discussed in public, studied and the subject of many national policies. But from the perspective of densely populated Europe, it is not entirely clear what exactly is going on. Access to healthy food is to counteract obesity, which is increasing very quickly (now it is an epidemic of obesity). Although some believe that the whole problem is artificial and blown, and the implemented policies are the result of efficient lobbying.
The food swamp describes the environment typical of the North American food system, which is corporate, industrial and increasingly global. Many of us, without realizing it, live in food swamps.
In my opinion, the elimination of food deserts is a real challenge in managing population and its health. I also hope that the results of this study will have an impact on zoning policies to reduce the damage associated with food swamps.
Author Response
Reviewer 3
Food deserts are characteristic of the United States and for several years has become a social problem discussed in public, studied and the subject of many national policies. But from the perspective of densely populated Europe, it is not entirely clear what exactly is going on. Access to healthy food is to counteract obesity, which is increasing very quickly (now it is an epidemic of obesity). Although some believe that the whole problem is artificial and blown, and the implemented policies are the result of efficient lobbying.
The food swamp describes the environment typical of the North American food system, which is corporate, industrial and increasingly global. Many of us, without realizing it, live in food swamps.
In my opinion, the elimination of food deserts is a real challenge in managing population and its health. I also hope that the results of this study will have an impact on zoning policies to reduce the damage associated with food swamps.
Response: Thank you for your comments.
Reviewer 4 Report
The research topic “Development of a method to locate deserts and food swamps following the experience of a region in Quebec, Canada” is interesting but cohesion lacks in the presentation. The manuscript can be considered for publication if revised all the comments.

Author Response
Reviewer 4
The research topic “Development of a method to locate deserts and food swamps following the experience of a region in Quebec, Canada” is interesting but cohesion lacks in the presentation. The manuscript can be considered for publication if revised all the comments.
Response: Thank you for your comments.
In page 2. Line 54. You have mentioned “socioeconomically disadvantaged neighborhood”. You need to clarify it. It is difficult to understand.
Response: We have adjusted our text and added two references. Several studies have shown that the availability and accessibility of healthy food can vary according to the income and educational level of individuals living in an area
- In page 2. Line 67. Please, rephrase the sentence and provide the reference.
Response: We have adjusted our text and added the reference.
- ln page 2. Line 68. You have mentioned the result of regression analyses. I could not find any regression analyses done. Is it a reference?
Response: We have adjusted our text and added the reference.
- Please provide reference for line 72, 97, 105, 170, 208, 262 and 274.
Response: References were added for lines 97, 208,262. For the other lines we do not understand the comment as references were already included for lines 72,105, 170, 274.
- In page 2. Line 85 to 87. Please clarify data collecting methods.
Response: We made the corrections.
- In page 3. Line 138. Make field validation process clearer and mention the reason for adding food stores and deleting convenience stores.
Response: We have adjusted our text and added more information.
- In page 5 and 7. Line 183. RFEI needs to be discussed more specially mean level of RFEI, Table 2. Check total of mean level in Table 2.
Response: We have adjusted our text and added more information.
- In page 7. Line 232. 4.2 must be 4.3.
Response: Change was done.
- In page 8. The author’s idea to improve food security and increase access to healthy food is lacking.
Response: Thank you for your comments. The objective of this study was to develop a method to assist regional authorities in their intervention measures. Consequently, we included in our discussion possible interventions and the interventions that Gaspesie put into place Lines 274 to 291.
Round 2
Reviewer 4 Report
You have put your effort to revise this manuscript entitled “Development of a method to locate deserts and food swamps following the experience of a region in Quebec, Canada”. However, comment 8 has not been addressed yet. The author added the reference in line 170 (comment 4) in the revised manuscript. It was not previously given in your original manuscript. I suggest author’s idea to improve food security and increase access to healthy food in discussion.
Author Response
You have put your effort to revise this manuscript entitled “Development of a method to locate deserts and food swamps following the experience of a region in Quebec, Canada”. However, comment 8 has not been addressed yet. The author added the reference in line 170 (comment 4) in the revised manuscript. It was not previously given in your original manuscript. I suggest author’s idea to improve food security and increase access to healthy food in discussion.
Response: Thank you for your comments.
- In page 8. The author’s idea to improve food security and increase access to healthy food is lacking.
Response: We have added a few paragraphs to illustrate ways to improve food security and access in our discussion. We also included the interventions done in the study region, Gaspesie, to increase access and reduce food insecurity.